# Influence of Large-Aperture Output Wavefront Distribution on Focal Spot in High-Power Laser Facility

**Jiamei Li [1,2], Hui Yu [1,2], Dawei Li [1], Li Wang [1], Junyong Zhang [1], Qiong Zhou [1], Fengnian Lv [1] and Xingqiang Lu [1,\*]**

1   Key Laboratory of High Power Laser and Physics, Shanghai Institute of Optics and Fine Mechanics, Chinese Academy of Sciences, Shanghai 201800, China
2   Center of Materials Science and Optoelectronics Engineering, University of Chinese Academy of Sciences, Beijing 100049, China
\*   Correspondence: xingqianglu@siom.ac.cn; Tel.: +86-021-6991-8292

**Abstract:** To improve the focal spot quality, the output wavefront of the Shenguang-II Upgrade facility is divided into four types based on the spatial frequency and division band of power spectral density. The influence of each on the focal spot was quantitatively studied, and the results indicate that the spatial profile, energy concentration, and peak intensity of the focal spot are mainly affected by low-spatial-frequency, and the relative intensity of the sidelobes is greatly affected by both the low- and mid-spatial counterparts. The peak-to-valley value of the wavefront of the Shenguang-II Upgrade should not exceed $2.27\lambda$ under the requirement that 50% of the energy is enclosed within 3 times the diffraction limit (DL), and it should be less than $2.45\lambda$ under the requirement that 95% is within 10DL. Meanwhile, the Strehl ratio dropped to within 0.2 under these conditions. These results can be applied to focal quality improvement in designing an adaptive optical system, optical element processing, as well as focal spot prediction in high-power laser facilities.

**Keywords:** transmitted wavefront; focal spot quality; Strehl ratio; energy concentration; relative intensity of sidelobes





## 1. Introduction

For high-power laser facilities, the wavefront distribution is an important factor that affects the laser driver's focal spot characteristics, which is severely affected by the transmitted wavefronts of the total optics of whole systems. In many high-power laser facilities, such as the National Ignition Facility (NIF) [1–4], Laser Megajoule (LMJ) [5–7], and Shenguang-II Upgrade (SG-II UP) [8–10] and Shenguang-III (SG-III) [11], pulses propagate through the multipass optical paths of booster amplifiers and cavity amplifiers, enter the final optics assembly (FOA), and finally shoot on the target. Many large-aperture optical elements exist in the complex chains of a facility, such as Nd:glass slabs, spatial filter systems, focusing elements in FOA, and debris shields. They all have different transmitted wavefronts because of the machining precision, pressure distortion during installation, and the effect of the environment. The lasers inevitably carry wavefront errors when propagating through the system. As a result, they affect the quality of the focal spot and decrease the focal energy concentration and focusing power density. This is unfavorable for physics experiments with high-power lasers.

The focal spot quality of laser drivers is mainly determined by the output wavefront error introduced by large-diameter optics [12]. In these facilities, the output beam, although being an input with a uniform wavefront, will have wavefront errors with spatial wavelengths ranging from a few centimeters to several hundred nanometers [13,14]. The wavefront peak-to-valley (PV) value, wavefront root-mean-squared gradient (GRMS), and power spectral density (PSD) have been used to evaluate beam wavefront errors. The spatial frequency [11–16] is divided into four types: low spatial frequency ($v < 0.0303$ mm$^{-1}$), mid-1 spatial frequency (PSD-1:0.0303–0.4 mm$^{-1}$), mid-2 spatial frequency (PSD-2:0.4–8.33 mm$^{-1}$),

and high spatial frequency $(v > 8.33\ \text{mm}^{-1})$. Among them, a low-spatial-frequency wavefront error is mainly affected by the spherical aberration, coma, and astigmatism of optics, which is usually analyzed via GRMS. A mid-spatial-frequency wavefront error, also called a waviness error, can easily induce intensity modulation and self-focusing effects and damage the elements during transmission. PSD is effective for analyzing mid-spatial-frequency wavefront errors. A high-spatial-frequency wavefront error mainly refers to roughness, which is usually characterized via the roughness root mean square (RMS) value and is often filtered out by the spatial filter system. The wavefront PV value is typically used to roughly characterize the beam wavefront errors.

Wavefront errors can degenerate the quality of the focal spot, and several reports have studied the influence of wavefront errors on the far-field characteristics of high-power laser drivers [11–17]. However, most of them only give the element machining requirements via a wavefront error for a single optical element. Few have considered systematic accumulated wavefronts with respect to the whole layout of a facility. The output wavefront in the FOA is difficult to simulate precisely because of various factors such as installation pressure by clamping, thermal distortion by pumping, and the deformable mirror in the adaptive optical (AO) system. In this paper, we investigated wavefronts of large-diameter beams, which were predicted by realistic transmitted wavefronts of Nd:glass slabs, debris shields, and the systematic accumulated static wavefront. Based on the layout of the SG-II UP facility, the wavefronts were divided into different spatial frequency types by PSD wavefront frequency bands, and the influence of each type on the focal spot characteristics was quantitatively analyzed. The results are expected to provide the allowable PV value range of the wavefront errors, which can be used to improve the quality of focal spots by the AO system and guide the processing of large-aperture optics. In addition, for high-power laser facilities, a focal spot on a target is difficult to measure due to the extremely high power density. This study is also beneficial to the prediction of focal spots according to the acquired data of the wavefront.

A wavefront error will cause the modulation of the near and far fields of a beam during the propagation, meaning the characteristics of the focal spot will be affected. Section 2 of this paper presents the theoretical model, the layout of the SG-II UP facility, and machine surface characteristics of large-diameter optics. The transmitted wavefront is given by the machine surface characteristics and is divided by the PSD band. Section 3 investigates the influence on the focal spot, focal shape, energy concentration, Strehl ratio, and relative intensity of sidelobes. Accordingly, the allowable wavefront PV value of the optics and the direction of optimization are provided. Finally, a conclusion is drawn in Section 4.

## 2. Theoretical Model

The layout of the SG-II UP facility is shown in Figure 1. The beam is injected into the main amplification chain by a pre-amplifier, propagates through a two-pass booster amplifier and four-pass cavity amplifier, and finally enters the FOA. In the SG-II UP facility, the booster amplifier and cavity amplifier are composed of five and eight pieces of Nd:glass slabs, respectively. After the beam enters the amplification system, it will pass through 42 Nd:glass slabs and at least one debris shield, and then shoot at the target. Additionally, the Hartmann sensor is placed between the output of the transport spatial filter and the injection of FOA to measure the beam wavefront for the correction of the adaptive optics system.

The actual output wavefront in the FOA is difficult to simulate accurately. Additionally, it can only be predicted with the static wavefront of the elements measured by an interferometer or AO system. To deduce a beam wavefront via the optics-transmitted wavefront, fortunately, the processing wavefront of the elements is often similar, and the wavefront errors introduced by the installation and pump are mainly low-spatial-frequency errors. Therefore, when predicting the output laser wavefront in the structure of Figure 1, two methods were selected in this study: one relies on the transmitted wavefronts of optics

which has high similarity, such as Nd:glass slabs and debris shields, and the other relies on accumulating the static wavefronts of optics in the chain of the SG-II UP facility.

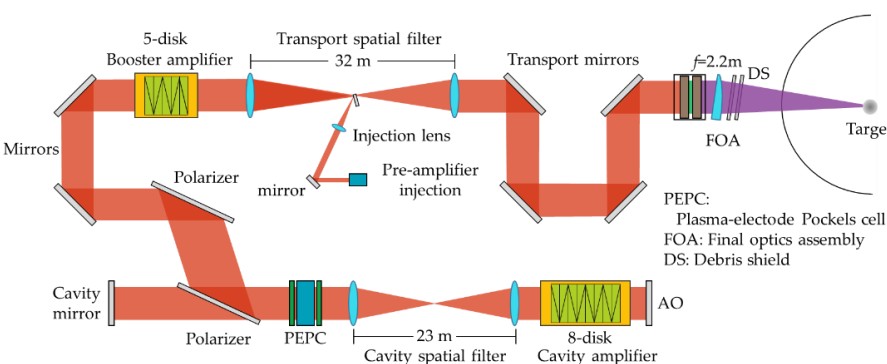

**Figure 1.** Layout of the SG-II UP facility.

Generally, the average PV of a Nd:glass wavefront is approximately $0.25\lambda$. A measured transmitted wavefront distribution of Nd:glass, which changes slowly without obvious periodic modulation, is displayed in Figure 2a and numbered I. The SG-II UP facility consists of eight beamlets, and the one we chose in our study contains 13 Nd:glass slabs, and the PV values of them are $0.25\lambda$, $0.25\lambda$, $0.21\lambda$, $0.26\lambda$, $0.28\lambda$, $0.30\lambda$, $0.27\lambda$, $0.38\lambda$, $0.22\lambda$, $0.65\lambda$, $0.14\lambda$, $0.24\lambda$, and $0.31\lambda$.

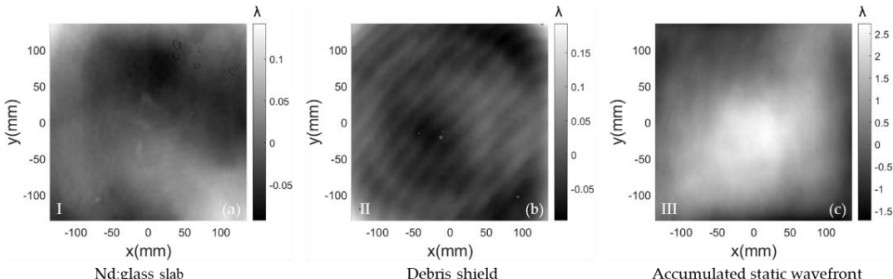

**Figure 2.** Typical wavefronts. (**a–c**) represent the measured wavefront of Nd:glass slab, debris shield, and an accumulated wavefront by superposing the wavefronts of all the elements according to the SG-II UP facility layout, respectively.

The transmitted wavefront PV of the debris shield in SG-II UP is generally between $0.3\lambda$ and $0.5\lambda$. A measured transmitted wavefront is shown in Figure 2b and numbered II, which carries rings with periodic stripe phase modulation.

An accumulated static wavefront, shown in Figure 2c and numbered III, was achieved by superposing all the measured transmitted wavefronts of the large-aperture elements according to the layout of the SG-II UP facility. The PV value reached $4.45\lambda$. The accumulated wavefront was smoother, and the stripes introduced by a single debris shield were inconspicuous.

We used these transmitted wavefronts and accumulated wavefronts as the predicted ones of the beam to analyze the influence of these three types of wavefronts on the focal spot.

The Holaser software [18,19], a self-developed software and formerly called the Laser designer, was used in the theoretical calculations. It is employed for the design and improvement of SG-II UP systems, and its validity has been proven experimentally. The software used the beam diffraction propagation method based on physical field tracing to calculate the output focal spot. The calculation formula used was as follows:

$$\nabla_\perp^2 E + 2ik_0 \frac{\partial E}{\partial z} + 2k_0^2 \frac{n_2}{n}|E|^2 E = 0 \tag{1}$$

where $n_0$ and $n_2$ are the linear refractive index and nonlinear refractive coefficient of the medium, respectively, and $k_0$ is the wavenumber $k_0 = n_0\omega_0/c$. The first two terms represent the diffraction effect, and the third represents the nonlinear self-focusing effect. Equation (1) can be solved using the Split-step Fourier method [20]. To study the influence of the wavefront on the focal spot, the nonlinear transmission is not analyzed in the calculation.

In addition, to improve the resolution of the focal spot calculation, the Fresnel diffraction we used was:

$$E(x,y) = \frac{\exp(ikz)}{i\lambda z} \iint E(x_0, y_0) \exp\left\{ \frac{ik}{2z}[(x-x_0)^2 + (y-y_0)^2] \right\} dx_0 dy_0 \tag{2}$$

where $(x_0, y_0)$ and $(x, y)$ are the points on the diffraction and observation screens, respectively.

The output of the SG-II UP system was a square beam with dimensions of 310 mm $\times$ 310 mm. In our calculation, it was assumed to be an ideal super-Gaussian profile, described as:

$$E(x,y) = A_0 \exp\left\{ -\frac{\ln 2}{2}\left[ \left(\frac{x^2}{a_x^2}\right)^{n_x} + \left(\frac{y^2}{a_y^2}\right)^{n_y} \right] \right\} \exp[-i\phi(x,y)] \tag{3}$$

where $A_0$ is the signal amplitude; $n_x$ and $n_y$ represent the spatial distribution of the laser pulse in the $x$ and $y$ directions, respectively ($n_x$, $n_y$ >1 is a super-Gaussian beam); $a_x$ and $a_y$ are the half-width at half maximum for the spatial distribution of a pulse in the $x$- and $y$-directions, respectively; and $\phi(x,y)$ is the output beam wavefront, predicted based on the optical path arrangement and large-diameter optical components.

To evaluate the wavefront of a SG-II UP system with respect to the PSD, it is necessary to solve and filter the wavefront distribution of the output beam. In this study, the least squares method [21,22] was used, and the solution was obtained using the discrete cosine transform. The wavefront information in different bands was obtained via the filtered wavefront according to the PSD requirements.

The filtering process is performed as follows [12]: First, a Fourier transform on $\phi(x,y)$ is performed to obtain the wavefront spectrum $\Phi$:

$$\Phi(f_x, f_y) = \iint \phi(x,y) \exp\left\{ -j2\pi(f_x x + f_y y) \right\} dx dy \tag{4}$$

The wavefront spectrum $\Phi$ is then filtered by $f_{Filter}$, which is a low-pass function, a band-pass function, or a high-pass function in different calculation situations.

$$\Phi'(f_x, f_y) = \Phi(f_x, f_y) f_{Filter} \tag{5}$$

Finally, inverse Fourier transform is performed on $\Phi'(f_x, f_y)$ to obtain the filtered wavefront phase.

$$\phi'(x,y) = \iint \Phi(f_x, f_y) f_{Filter} \exp\left\{ j2\pi(f_x x + f_y y) \right\} df_x df_y \tag{6}$$

The 10th-order Gaussian distribution function was used as the filtering function $f_{Filter}$ in this study and was substituted into Equation (3). Equations (1) and (2) were then used to calculate along the propagation optical path of the FOA, and finally, the distribution of the focal spot of the large-diameter laser could be obtained.

The energy concentration (*Ec*), Strehl ratio (SR), and sidelobe relative intensity were used to evaluate the quality of the focal spot. *Ec* is described as [23]:

$$Ec = \int_0^b \int_0^{2\pi} |E(x,y)|^2 r\,dr\,d\theta \bigg/ \left[ \int_0^\infty \int_0^{2\pi} |E(x,y)|^2 r\,dr\,d\theta \right] \tag{7}$$

If *Ec* is larger, the energy on the focal spot is more concentrated, and the quality of the focal spot is better than that of the smaller value at the same focal spot size.

The Strehl ratio is the ratio of the actual far-field peak intensity along the beam axis to the peak light intensity of an ideal beam with the same amplitude distribution and a uniform phase, given by [24]:

$$SR = \frac{\left| \int A(x,y) \exp[i\phi(x,y)] dx dy \right|^2}{\left| \int A(x,y) dx dy \right|^2} \tag{8}$$

where $A(x,y)$ represents the actual amplitude distribution of the beam, and $\phi(x,y)$ is the phase distribution. Generally, a higher SR indicates a higher peak intensity for the focal spot.

The relative intensity of the focal spot sidelobes is defined as the ratio of the focal spot sidelobe intensity to the peak intensity. This is related to the power density of the focal sidelobe. The greater the relative intensity, the higher the possibility that hole-blocking occurs in the spatial filter system in the chain.

The transmitted wavefront is segmented by a PSD wavefront frequency band. By changing the wavefront PV value of the segmentation, its effect on the focal spot could be evaluated in this study.

## 3. Numerical Calculation and Analysis Results

According to Section 2, the transmitted wavefront of the Nd:glass, the debris shield, and static wavefront, recorded as wavefront I, II, and III, respectively, are typically used as the three typical wavefronts of the output beam in FOA. They are used to simulate the output wavefront of the facility as the original wavefronts. According to the PSD band, the original wavefronts are divided into low spatial frequency: $\nu < 0.0303$ mm$^{-1}$, PSD-1: 0.0303–0.4 mm$^{-1}$, PSD-2: 0.4–8.33 mm$^{-1}$; and high spatial frequency: $\nu > 8.33$ mm$^{-1}$; the filtered wavefronts are shown as depicted in Figure 3. The distributions of the low spatial frequency (column 1) and the PSD-1 (column 2) are different. However, the high spatial frequency (column 4) is limited by the resolution of the ZYGO interferometer, and it is mostly noise from the measurement process and will not represent the real high-spatial-frequency distribution; therefore, it is not discussed and analyzed in this paper.

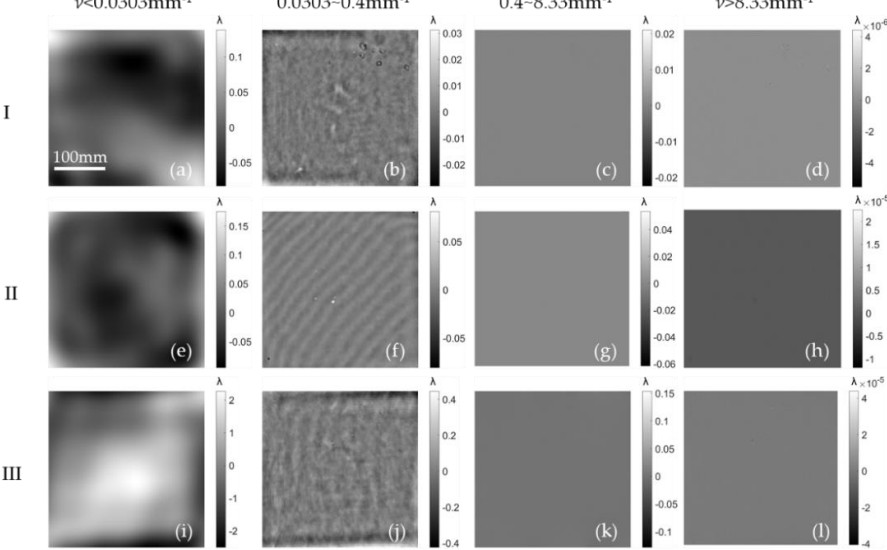

**Figure 3.** Filtered wavefronts with different spatial frequencies. (**a**–**d**) represent the wavefront distribution of the low spatial frequency, the PSD-1, the PSD-2, and high spatial frequency of type I, respectively; (**e**–**l**) represent the four wavefronts of type II and type III, respectively, same as (**a**–**d**). The scale bar is 100 mm, and all subplots are the same size.

We normalized the wavefront PV value. When the PV values of the original wavefronts are $1\lambda$, and the PV value ratio of each filtered wavefront part is maintained, the PV values of the filtered wavefront are listed in Table 1. From Table 1, for the PV values of the low-spatial-frequency type, the largest is wavefront III, while the lowest is II; for the mid-spatial-frequency type, the results are the opposite. This indicates that the accumulated wavefront, wavefront III, has certain complementarity in the mid-frequency types, which is beneficial for a more uniform and smoother wavefront distribution. The following section quantitatively analyzes the effect of different wavefront types on the focal spot characteristics by varying the PV values.

**Table 1.** PV values of filtered wavefront types when original wavefront PV = $1\lambda$.

| Wavefront Type | Low Spatial Frequency | PSD-1 Frequency | PSD-2 Frequency | High Spatial Frequency |
| --- | --- | --- | --- | --- |
| | <0.0303 mm$^{-1}$ | 0.0303–0.4 mm$^{-1}$ | 0.4–8.33 mm$^{-1}$ | >8.33 mm$^{-1}$ |
| I | $0.959\lambda$ | $0.258\lambda$ | $0.187\lambda$ | $4.287 \times 10^{-5}\lambda$ |
| II | $0.957\lambda$ | $0.559\lambda$ | $0.399\lambda$ | $1.194 \times 10^{-4}\lambda$ |
| III | $1\lambda$ | $0.190\lambda$ | $0.062\lambda$ | $1.846 \times 10^{-5}\lambda$ |

### 3.1. Shape of Focal Spot

The three original wavefronts in Figure 2 and the filtered wavefront of the low-spatial-frequency, PSD-1, and PSD-2 parts in Figure 3 were considered as the beam wavefront. The far-field focal spots of these wavefronts are shown in Figure 4, where all the PV values of the different spatial frequency types are equal to $2\lambda$ for an obvious comparison, which were calculated with the parameters of the FOA in the SG-II UP facility [25].

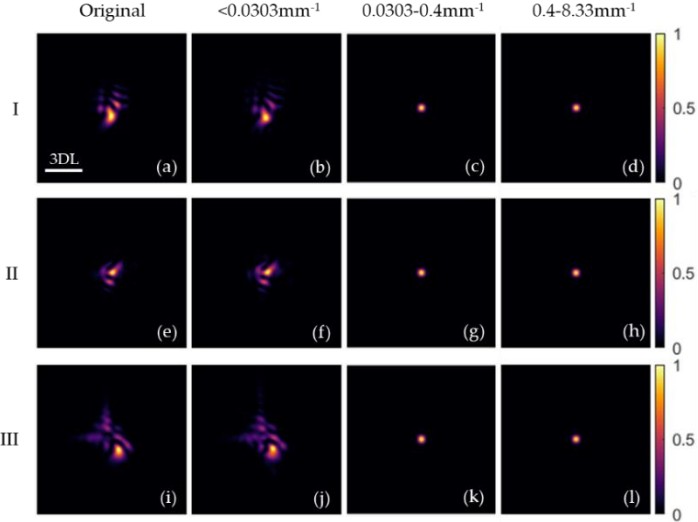

**Figure 4.** Focal spot shapes under different wavefront parts with PV = $2\lambda$. (**a–d**) represent the focal spots of the original, the low spatial frequency, PSD-1, and PSD-2 of type I, respectively, when the PV value of them all is $2\lambda$; (**e–l**) represent the focal spots of type II and type III, respectively, same as (**a–d**). 3DL is a scale bar, and all subplots are the same size.

In Figure 4, rows 1–3 present the focal spots corresponding to wavefronts I, II, and III, respectively. Column 1 shows the focal profiles of the unfiltered original wavefront distribution, and column 2 shows the focal profiles of the low-spatial-frequency wavefront. Columns 3 and 4 represent the focal profiles of PSD-1 and PSD-2, respectively. Additionally, 3DL is a scale bar, which means 3 times the diffraction limit (DL). The one-dimensional relative intensity distributions of the focal spot are shown in Figure 5, obtained by the focal profiles of the centers in Figure 4.

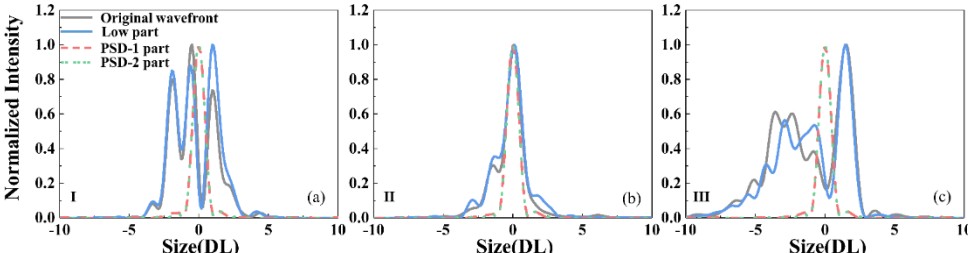

**Figure 5.** Normalized intensity of focal spot different wavefront types with PV = 2λ. (**a**) is the focal intensity of the original, the low spatial frequency, the PSD-1, and the PSD-2 of type I, when the PV value of them all is 2λ, respectively; (**b**,**c**) are the focal intensity of type II and type III, respectively, same as (**a**).

Two features can be obtained from Figures 4 and 5. First, when different wavefronts have the same PV value, the focal spot of the low spatial frequency is roughly similar to the original wavefront in the three cases, which has a similar profile and dispersion. Second, there is no apparent dispersion or splitting in the focal spot of the mid-spatial-frequency band. Therefore, it can be concluded that the focal profile is mainly determined by the low-spatial-frequency wavefront.

*3.2. Energy Concentration of Focal Spot*

Energy concentration (*Ec*) is an important parameter to evaluate the effecient energy of the focal spot in the field of high-power lasers. Moreover, the *Ec* is closely related to the dispersion of the focal spot. The relationship between *Ec* and the wavefront distribution is of great significance for facility optimization.

The low spatial frequency (<0.0303 mm$^{-1}$), low spatial frequency + PSD-1 (<0.4 mm$^{-1}$), low spatial frequency + PSD-1 + PSD-2 (<8.33 mm$^{-1}$), and ideal wavefront of I, II, and III were used as the calculation wavefronts of the beam, and the focal spot distribution was calculated to obtain the *Ec* of the focal spot, as shown in Figure 6, where the original wavefront PV = 2λ and the PV value ratio of each filtered wavefront was maintained and unchanged. Here, the gray, blue, red, and green curves represent the *Ec* of the low spatial frequency, low spatial frequency + PSD-1, low spatial frequency + PSD-1 + PSD-2, and the ideal beam without wavefront errors, respectively.

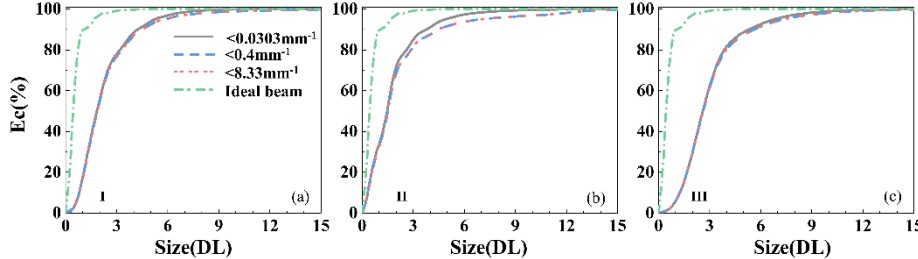

**Figure 6.** *Ec* of focal spots when the unfiltered wavefront PV = 2λ and the PV value ratio of each filtered wavefront is maintained. (**a**–**c**) are the *Ec* of type I, II, and III, respectively.

The gray, blue, and red curves have similar change trends, far worse than the green curve under the same focal size. The red and blue curves almost coincide. These results indicate that *Ec* is mainly determined by the low-spatial-frequency wavefront, and the PSD-2 has almost no effect on *Ec*. Therefore, to obtain a high *Ec* of the focal spot, the low-spatial-frequency wavefront error should be focused on during AO correction in the future.

In high-power laser facilities, energy concentration within 3 DL ($Ec_{3DL}$) and 10DL ($Ec_{10DL}$) is usually focused on when evaluating the quality of the focal spot. The $Ec_{3DL}/Ec_{10DL}$ variations in each spatial-frequency part with the original wavefront PV value are given

in Figure 7. Figure 7a–c display the changes in $Ec_{3DL}$ of wavefront I, II, and III, while Figure 7d–f present the changes in $Ec_{10DL}$. In each figure, the gray, blue, and red curves represent the $Ec$ of the low-spatial-frequency wavefront (<0.0303 mm$^{-1}$), the low-spatial-frequency + PSD-1 part (<0.4 mm$^{-1}$), and the low-spatial-frequency + PSD-1 + PSD-2 part (<8.33 mm$^{-1}$), respectively.

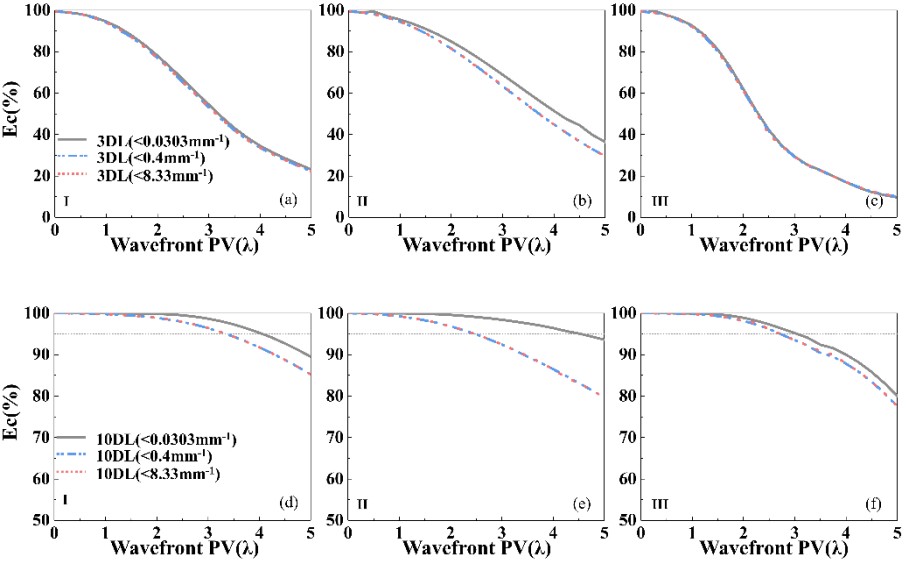

**Figure 7.** $Ec_{3DL}/Ec_{10DL}$ variations in each spatial-frequency type with the original wavefront PV value. (**a–c**) are the changes in wavefront I, II, and III on $Ec_{3DL}$, respectively; (**d–f**) are the changes of wavefront I, II, and III on $Ec_{10DL}$, respectively.

In Figure 7, the overlap of the curves between <0.4 mm$^{-1}$ and <8.33 mm$^{-1}$ indicates that $Ec$ is less affected by the PSD-2 wavefront. Additionally, the separation of the curves between <0.4 mm$^{-1}$ and <0.0303 mm$^{-1}$ indicates that PSD-1 has some influence on $Ec$. When the original wavefront PV reached $2\lambda$, the $Ec_{3DL}$ of wavefronts I, II, and III were calculated as 76.9%, 81.5%, and 61.3%, respectively, while the $Ec_{10DL}$ of wavefronts I, II, and III were 98.9%, 96.8%, and 98.2%, respectively. This demonstrates that wavefront III has a significant influence on $Ec_{3DL}$, while wavefront II has a significant influence on $Ec_{10DL}$.

In particular, if $Ec_{3DL} \geq 50\%$ or $Ec_{3DL} \geq 95\%$ is required in the SG-II UP facility, the maximum allowable PV values are provided in Table 2, according to Figure 7.

**Table 2.** Allowable PV values of original wavefront under $Ec_{3DL} = 50\%$ and $Ec_{10DL} = 95\%$.

| Energy Concentration | $Ec_{3DL} = 50\%$ | | $Ec_{10DL} = 95\%$ | |
|---|---|---|---|---|
| | Low Spatial Frequency | Full Spatial Frequency | Low Spatial Frequency | Full Spatial Frequency |
| I | $3.19\lambda$ | $3.14\lambda$ | $4.06\lambda$ | $3.35\lambda$ |
| II | $4.09\lambda$ | $3.71\lambda$ | $4.54\lambda$ | $2.45\lambda$ |
| III | $2.30\lambda$ | $2.27\lambda$ | $3.05\lambda$ | $2.72\lambda$ |

Table 2 shows the maximum allowable PV values of the original wavefront when $Ec_{3DL} = 50\%$ and $Ec_{10DL} = 95\%$ under low- and full-spatial-frequency wavefronts. For $Ec_{3DL} = 50\%$, the discrepancy of PV value between the low-spatial-frequency type and full spatial frequency is very small, which indicates the mid-spatial-frequency type has a weak influence. However, for $Ec_{10DL} = 95\%$, the effect of the mid-spatial-frequency wavefront is significant and cannot be neglected. Particularly, because of wavefront II with periodic modulation in PSD-1, as shown in Figure 3f, its discrepancy is up to $2.09\lambda$ when

$Ec_{10DL}$ = 95%. Therefore, to obtain high energy concentration, it is necessary to avoid a periodic wavefront error in the FOA beam.

In addition, as shown in Table 2, for the three types of wavefronts, the requirements of $Ec_{3DL}$ = 50% and $Ec_{10DL}$ = 95% are not equivalent. Wavefront III has the smallest differences in allowable PV values between the two requirements, whereas wavefront II has the largest. Furthermore, if there are no periodic modulations in the laser wavefront, the allowable PV value of the two requirements is generally less than $0.5\lambda$, and $Ec_{3DL} \geq 50\%$ is a more rigorous requirement.

For the three typical wavefronts in this study, if $Ec_{3DL} \geq 50\%$ or $Ec_{10DL} \geq 95\%$ is required, the wavefront PV value of the output laser in FOA must not exceed $2.27\lambda$ and $2.45\lambda$, respectively.

### 3.3. Strehl Ratio of Focal Spot

The wavefronts of different spatial-frequency types of I, II, and III were used as the calculated wavefronts, as in Section 3.2, and the variations in their SRs with the PV value of the original wavefront are shown in Figure 8.

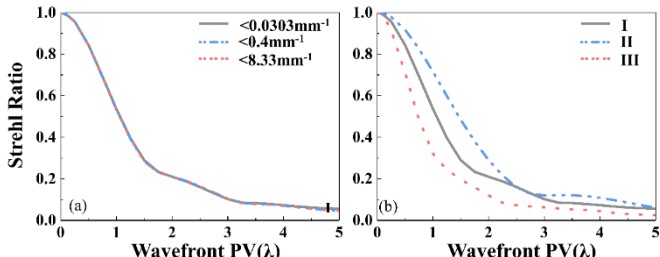

**Figure 8.** Variations in Strehl ratio with PV value of the original wavefront. (**a**) The SR of wavefront I with different spatial-frequency types. (**b**) The SRs of the low frequency of wavefront I, II, and III.

Figure 8a shows the SR variations in wavefront I with different spatial-frequency types, and the three SR curves almost overlap. This indicates that the SR is almost affected by the low spatial frequency. There are similar changes to the SRs of wavefront II and III.

Figure 8b compares the SRs of the wavefront I, II, and III under the low-spatial-frequency wavefront. From Figure 8b, the SR of wavefront III is the fastest to decline and has the lowest value, while that of wavefront II is the slowest to decline and has the highest value. This is because the low-spatial-frequency PV of wavefront III is the largest, while that of wavefront II is the smallest under the same original wavefront PV from Table 1. When $Ec_{3DL}$ = 50% and $Ec_{10DL}$ = 95%, the allowable PV values are $2.27\lambda$ and $2.45\lambda$, respectively, and their SRs drop to within 0.2.

According to the different requirements of focal spot SR, the allowable PV values of original wavefronts are listed in Table 3.

**Table 3.** Allowable PV values of original wavefront under SR requirements.

| SR | Wavefront I | Wavefront II | Wavefront III |
|---|---|---|---|
| 0.5 | $1.06\lambda$ | $1.46\lambda$ | $0.73\lambda$ |
| 0.25 | $1.68\lambda$ | $2.14\lambda$ | $1.23\lambda$ |
| 0.2 | $2.11\lambda$ | $2.32\lambda$ | $1.48\lambda$ |
| 0.1 | $3.03\lambda$ | $4.19\lambda$ | $2.14\lambda$ |

### 3.4. Relative Intensity of Focal Sidelobes

When analyzing the quality and the peak intensity of the focal spot, it is necessary to pay attention to the intensity distribution of the focal sidelobes. The higher the sidelobe intensity, the higher the hole-blocking risk during laser propagation in the amplification chain. The three types wavefronts have similar effects on sidelobe intensity. The influence

of different filtered types of wavefront I on the sidelobe intensity is shown in Figure 9 when the unfiltered original wavefront PV = 2$\lambda$.

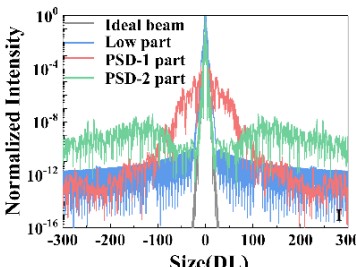

**Figure 9.** Focal spot intensity of different filtered parts of wavefront I when unfiltered wavefront PV = 2$\lambda$.

From Figure 9, it can be observed that different spatial-frequency types have different effects on the focal spot. The low-spatial-frequency type mainly affects the intensity distribution near the focal mainlobe. The PSD-1 type mainly affects low-level sidelobes, and PSD-2 mainly affects high-level sidelobes. From the diffraction divergent angle $\theta = \lambda / \Lambda$, the corresponding focal spot size is $r = (D/\Lambda) \cdot DL$, where $\lambda$ is the wavelength, $\Lambda$ is the spatial period, and $D$ is the beam aperture. Therefore, the low-spatial-frequency type, PSD-1, and PSD-2 mainly affected the focal spot within approximately 10DL, 10DL-124DL, and larger than 124DL, respectively. In addition, the relative intensity of the focal sidelobes under the low spatial frequency, PSD-1, and PSD-2, are about >$10^{-4}$, >$10^{-9}$, and >$10^{-12}$ when the original wavefront PV = 2$\lambda$, respectively.

The relative intensity of the PSD-2 frequency starts to increase around 50DL, as shown in Figure 9, and the divergent angle is approximately 170 µrad, which is close to the angle of the aperture radius. To prevent the hole-blocking effect caused by the high energy density of the sidelobe, the intensity of the focal spot at 50DL should not be too high. The relative intensities of the three wavefronts at 50DL are studied, and the variations with the PV value of original wavefront are shown in Figure 10.

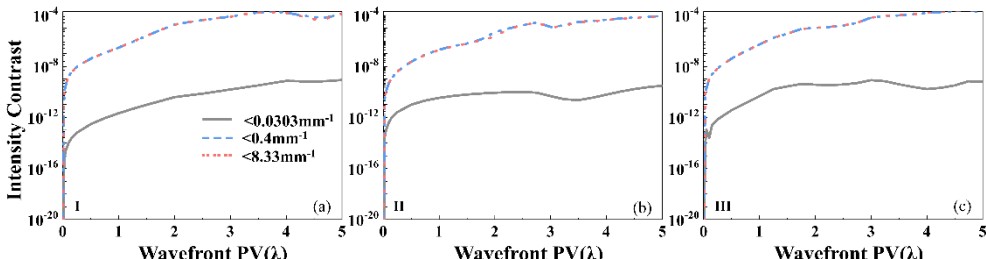

**Figure 10.** Variation in the relative intensity of focal sidelobes with PV value under different spatial frequency wavefronts. (**a**–**c**) are the focal relative intensity at 50DL of wavefront I, II, and III, respectively.

Figure 10 depicts the effect of the PV value on the focal sidelobe intensity with the different spatial-frequency types. In this figure, the separation of the gray and blue curves and the coincidence of the blue and red curves indicate that the relative intensity of 50DL is mainly affected by the low spatial frequency and PSD-1 frequency, while is almost not influenced by the PSD-2 frequency. With the increase in the original wavefront PV, the relative intensity of the sidelobes rises. When PV = 2$\lambda$, the relative intensity at 50DL deteriorates to $10^{-6}$–$10^{-5}$. If the relative intensity is required to be less than $10^{-8}$, the allowable PV values of wavefronts I, II, and III should not exceed 0.35$\lambda$, 0.37$\lambda$, and 0.37$\lambda$, respectively.

## 4. Conclusions

In this study, we predict the output wavefronts via the transmitted wavefront of optics and analyze the wavefront effect on the focal spot. It is found that the focal spot shape, energy concentration, and SR are determined by the low-spatial-frequency wavefront, while focal sidelobes are affected by mid spatial frequency. These rules are also applicable to other beams of different wavelengths. From the analysis of different typical wavefronts in this paper, the wavefront with periodic distortions has a significant effect on $Ec$. Moreover, the requirement that $Ec_{3DL} = 50\%$ is stricter than that for $Ec_{10DL} = 95\%$ when the wavefront is without periodic modulation. If SR > 0.1 is needed, the PV value should not exceed $2.14\lambda$. When PV = $2\lambda$, the relative intensity of 50DL deteriorates to $10^{-5}$. In high-power laser facilities, the wavefront errors of the low spatial frequency should be focused on, and they should be reduced during optical element processing and corrected by the AO system based on the allowable PV values during the transmission.

**Author Contributions:** Conceptualization, J.L. and X.L.; methodology, J.L. and X.L.; software, X.L.; validation, H.Y., D.L. and L.W.; formal analysis, J.L.; investigation, H.Y., L.W. and Q.Z.; data curation, J.L., J.Z. and X.L.; writing—original draft preparation, J.L.; writing—review and editing, H.Y., D.L., J.Z. and F.L.; supervision, Q.Z. and F.L.; project administration, X.L.; funding acquisition, J.Z. and X.L. All authors have read and agreed to the published version of the manuscript.

**Funding:** This research was funded by the National Natural Science Foundation of China, grant number 62175245.

**Institutional Review Board Statement:** Not applicable.

**Informed Consent Statement:** Not applicable.

**Data Availability Statement:** All the data reported in this paper are presented in the main text. Any other data will be provided on request.

**Conflicts of Interest:** The authors declare no conflict of interest. The funders had no role in the design of the study; in the collection, analyses, or interpretation of data; in the writing of the manuscript; or in the decision to publish the results.

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
