# Peer review of "Influence of Large-Aperture Output Wavefront Distribution on Focal Spot in High-Power Laser Facility"

_photonics, doi:10.3390/photonics10030270_

Round 1
Reviewer 1 Report (New Reviewer)
Notes are in the file.

Author Response
Please see the attachment.

Reviewer 2 Report (New Reviewer)
The paper presents a thorough and well-conducted investigation into the relationship between wavefront distribution and focal spot quality in the Shenguang-II Upgrade facility. The authors provide an impressive amount of data, and their detailed analysis is highly appreciated.
The methodology employed by the authors is robust, and the results of their investigation provide valuable insights into the complex relationship between wavefront distribution and focal spot quality. The study concludes that the focal spot shape, energy concentration, and Strehl ratio are primarily determined by the low-spatial-frequency wavefront, while the focal sidelobes are affected by mid-spatial-frequency parts. The authors also provide specific constraints on the wavefront peak-to-valley value for achieving optimal focal spot quality.
The conclusions drawn from the data and analysis are clear and well-supported. The focus on the reduction of low-spatial-frequency wavefront distortions during optical element processing and correction by adaptive optical systems is particularly noteworthy.
In conclusion, this article provides a comprehensive investigation into the influence of the wavefront distribution on the focal spot quality in high-power laser facilities. The authors' attention to detail, the robust methodology employed, and the thorough analysis make this a valuable contribution to the field. The results of this study have the potential to inform the design and optimization of high-power laser facilities.
Author Response
Thanks for the Reviewer’s positive comments. We have tried our best to revise our manuscript.
Reviewer 3 Report (New Reviewer)
The authors studied the effects of the wavefront distorsions on the focal spot of the Shenguang-II Upgrade facility, and give the results of different types of wavefront that have different effects. this work can be applied in high power laser facilities to control or improve the the focal spot. It is useful in the high power laser design. I, substantially, agree to accept this paper to be published on photonics after the revision.
please the anthors refer to the file I have uploaded.
1. some specilized words in laser field have been corrected.
2. lots of linguistic errors have been corrected. So many that not listed here, but still strongly recommended that the paper be language polished by native english speaker.
3. all figures and tables in the paper are appeared very sudden and then followed the disscusions, how the figs and tables are draw. please introduce.

Round 2
Reviewer 1 Report (New Reviewer)
The manuscript may be recommended for publication.
This manuscript is a resubmission of an earlier submission. The following is a list of the peer review reports and author responses from that submission.
Round 1
Reviewer 1 Report
Li et al. present the effects of the output wavefront on the focal spot of high-power lasers. The authors provide the simulation results to show how low-spatial-frequency and mid-spatial-frequency wavefronts determine different properties of the beam shape. This manuscript, however, can be improved if the authors can provide following information:
Major comments:
1. Is Fig. 2 the realistic beam shape or the simulated pattern? If it is simulated pattern, author should try to provide a realistic beam pattern of SG-II UP family for the over evaluation of the theoretical approach. If it is a realistic beam pattern, author should state clearly in the manuscript.
2. The caption Fig.2 is poorly presented. The caption needs more details to describe figure a, b, and c. There should be the label for y axis.
3. Fig. 3 needs to be reworked. First, the scale bars are required to better evaluate the patterns of images. Again, authors need to put more information in caption part. Like (a-d) represent, ……
4. The quality of Fig. 5 is very low. It is hard to distinguish PSD-1 and PSD-2 parts, although they are very similar. The legend of the figure should be placed properly. Authors can try to change the y axis from 0 to 1 to 0 to 1.2 which will allow some space on the top of the figure. Again, caption needs more details.
5. Since the author mentioned their method can be applied to other wavelengths. It is recommended that the author should briefly mention/discuss the potential difficulties when individuals use their method to improve the beam pattern in a real experiment.
Minor comments:
1. Author need to provide the full name of “PV” in the text.
2. Line 89. Change “the Fig. 1” to “Fig. 1”.
3. Line 190. “Table 1. This is a table. Tables should be placed in the main text near to the first time they are cited.” I believe this sentence should be reworked or removed. A proper title should be made for Table 1.
4. Label the scale bars in Fig. 4.
5. In Fig. 6, 7, 8, 9, and 10, there is no black line, only gray line.
Reviewer 2 Report
Unfortunately, I cannot recommend the manuscript for publication in its current form.
The manuscript is rather carelessly prepared. For example, it is difficult to understand which data given in the manuscript are experimental and which are calculated. It is even difficult to understand the parameters of the spectral intervals considered in the manuscript, since there is no caption in the table where they are presented. The motivation for the work and the main results are not clearly stated. It seems that the conclusions of the work are based on erroneous statements and do not inspire confidence.
The following is an incomplete list of comments, which, however, are sufficient to draw conclusions about the quality of the preparation of the manuscript:
А number of abbreviations (such as PV and AO) is not defined in the manuscript.
Fig. 2 does not have grayscales.
Line 169 - Fig. 2 instead of Fig. 3.
Caption of Fig. 5 is very brief and makes it difficult to understand the content of the figure.
line 157: Formula (8) seems to be not consistent with the standard definition of SR.
line 180: Figure 3 do not have grayscales. Panels (c)(d)(g)(h)(k)(l) are not informative.
line 190 Table 1 - no caption. What is “4.287×10-05λ”?
line 195 The statement “calculated by the FOA arrangement of the SG-II facility” is not clear.
line 197: Figure 4 is not clear. Shouldn't column 1 be the sum of columns 2, 3, and 4? The differences between 1 and 2 are clearly visible and should be shown in columns 3 and 4.
lines 207-213: The conclusion that high spatial harmonics do not affect (or only slightly affect) the focal brightness seems to be rather strange. It usually turns out that spatial components with the same amplitude affect the focal brightness the stronger, the higher the spatial frequency (for example, forming a wide halo in the focal plane or in some other way, depending on the spatial frequency). Thus, this part of the conclusions of the manuscript looks strange and requires at least some additional explanations.
In addition, the practical results of these studies are not stated. Indeed, adaptive optics has limitations in terms of its ability to compensate phase distortions with high spatial frequencies, but what the authors would recommend in terms of the use of adaptive optics based on their results?
Round 2
Reviewer 2 Report
The authors did not answer my main question in the corrected version of the manuscript. Hence I will try to reformulate it in a different way.
In fact, the authors use some real wavefronts, which they filter using different spatial frequency filters. Then the authors claim: "The three original wavefront distributions in Figure 2 and filtered wavefront of the low-frequency, PSD-1, and PSD-2 parts in Figure 3 were considered as the beam wave front. When all the PV values of these wavefronts reached 2λ, their far-field focal spots appeared as shown in Figure 4, which were calculated with the parameters of the FOA in 210 SG-II UP facility [25]". Does it mean that PV of the original wavefront is set to 2λ, the low-frequency wave front is set to 2λ, the PSD-1 wavefront is set to 2λ etc.? If so, than the results presented in Fig.3 look rather strange as the higher-spatial-frequencies-filtered wavefronts (PSD-1 and PSD-2) are focused into ideal focal spots. Could the authors comment this point?
Additionally, I draw the attention of the authors that the grayscale in Figures 2 and 3 is designated as PV / lambda, but it seems that this is not PV, but the phase front itself, measured in lambdas.
Round 3
Reviewer 2 Report
Reading each new version of the article, I inline to trust the results less and less.
Indeed, the authors demonstrate that wave fronts with the same amplitude of spatial inhomogeneities, but with different spatial frequencies of these inhomogeneities, are focused differently. Moreover, according to the results obtained by the authors, wave fronts with large spatial frequencies of inhomogeneities are focused better than wave fronts with lower spatial frequencies of inhomogeneities, which contradicts physical intuition.
As far as I understand the procedure used by the authors, the possible error begins with formula 4.
In order to get the wave front only from a part of the spatial spectrum, one need to act differently, not like in the article.
The authors apply Fourier transform to the wavefront (just to the spatial phase (!)), impose a spatial frequency filter on it and take the inverse Fourier transform, hence moving back into x,y space. They claim that as a result they have obtained a wavefront corresponding to this spectral band, which is not true.
The correct way to do this is:
- Fourier transform must be taken from the entire wave field (its amplitude together with its phase);
- Apply a spatial spectral filter to the result;
- Take the inverse Fourier from this result and get a new field function in x,y coordinates;
- Take an argument from the obtained field and it will be a wavefront in x,y coordinates;
Since the very procedure for obtaining a wave front for a spatial frequency band is incorrect, then further results are doubtful ...
In my earlier reviews, I assumed that the authors did not state the idea clearly enough, but their latest corrections to the text of the article convinced me otherwise.
As a result, I cannot recommend the article for publication in this form.
Author Response
Dear Editors:
On behalf of my co-authors, we thank you very much for giving us an opportunity to submit our manuscript, we appreciate the editor and reviewers very much for their positive and constructive comments and suggestions on our manuscript entitled “Influence of Large-aperture Output Wavefront Distribution on Focal Spot in High-power Laser Facility”. (Manuscript ID: photonics-1879432).
We have studied the 3rd-round comments from Reviewer 2 and checked the theory carefully. We have found that the conclusions of Reviewer 2 may be unscientific. On this occasion, I merely wish to ask for an opportunity to explain it, and the reasons as follows:
1. The view of Reviewer 2 on wavefront filtering seems inconsistent with the paper reports. When filtering the wavefront, the Fourier transform is performed on the wavefront while is not on the entire wave field (its amplitude together with its phase)””, as the power spectral density (PSD) of wavefront in Reference 1 and its screenshot as follows.
Many analyses of wavefront distortion, such as PSD[2-3] and wavefront root-mean-squared (RMS) gradient[4], are calculated based on the wavefront. In our manuscript, the filtered wavefront is considered the new phase distribution of the wave field, and the focal spot is calculated by the wave field, including the amplitude and the new phase.
2. There may also be a few discrepancies in the understanding of Reviewer 2 about the spatial frequency of wavefront in this manuscript, as Reference 5.
We would like to express our great appreciation to you and reviewers for comments on our paper, and give us an opportunity to explain it. Looking forward to hearing from you.
Thank you and best regards.
Yours sincerely,
Please address all correspondence concerning this manuscript to me at xingqianglu@siom.ac.cn or jmli@siom.ac.cn.
Xingqiang Lu, Jiamei Li
Reference
[1] David M. Aikens, C. Robert Wolfe, and Janice K. Lawson "Use of power spectral density (PSD) functions in specifying optics for the National Ignition Facility", Proc. SPIE 2576, International Conference on Optical Fabrication and Testing, (2 August 1995);
https://doi.org/10.1117/12.215604
[2] Janice K. Lawson, David M. Aikens, R. Edward English Jr., and C. Robert Wolfe "Power spectral density specifications for high-power laser systems", Proc. SPIE 2775, Specification, Production, and Testing of Optical Components and Systems, (19 August 1996);
https://doi.org/10.1117/12.246761
[3] Helmut H. Toebben, Gabriele A. Ringel, Frank Kratz, and Dirk-Roger Schmitt "Use of power spectral density (PSD) to specify optical surfaces", Proc. SPIE 2775, Specification, Production, and Testing of Optical Components and Systems, (19 August 1996);
https://doi.org/10.1117/12.246751
[4] Janice K. Lawson, Jerome M. Auerbach, R. Edward English Jr., Mark A. Henesian, John T. Hunt, Richard A. Sacks, John B. Trenholme, Wade H. Williams, Milton J. Shoup III, John H. Kelly, and Christopher T. Cotton "NIF optical specifications: the importance of the RMS gradient", Proc. SPIE 3492, Third International Conference on Solid State Lasers for Application to Inertial Confinement Fusion, (23 July 1999);
https://doi.org/10.1117/12.354145
[5] Jean M. Bennett, Lars Mattsson “Introduction to Surface Roughness and Scattering”, Optical Society of America, (1989), pp. 4 & 48-50;
https://dx.doi.org/10.1364/1557521085
